# Impact of Staining Methods and Human Factors on Accuracy of Manual Reticulocyte Enumeration

**DOI:** 10.3390/diagnostics12092154

**Published:** 2022-09-05

**Authors:** Lin-Lin Pan, Hsiu-Chen Yu, Ching-Hui Lee, Kuo-Chuan Hung, I-Ting Tsai, Cheuk-Kwan Sun

**Affiliations:** 1Department of Laboratory Medicine, Chang Gung Memorial Hospital, Chiayi 61363, Taiwan; 2Department of Anesthesiology, Chi Mei Medical Center, Tainan 71004, Taiwan; 3Department of Emergency Medicine, E-Da Hospital, Kaohsiung 82445, Taiwan; 4School of Medicine, College of Medicine, I-Shou University, Kaohsiung 82445, Taiwan

**Keywords:** reticulocyte count, hematological disease, erythropoiesis, anemia

## Abstract

Although peripheral blood reticulocyte enumeration reflects bone marrow functional integrity, which is important for differential diagnosis of hematological diseases, the factors affecting its accuracy have not been adequately addressed. Using 100 consecutive venous blood samples being processed with four supravital staining techniques [i.e., brilliant cresyl blue (BCB), new methylene blue (NMB), and BCB/NMB with Liu’s stain] for reticulocyte enumeration, two technologists (senior vs. junior) conducted microscopic counting. The results were compared with those obtained with an automated system (Sysmex XE-5000) that served as the standard. The aims of this study were to identify (1) the technique that gave the most reliable outcome, and (2) possible human factors (i.e., seniority, repeated counting) that may affect the counting results. Analysis showed least bias (i.e., deviation from automated counting) associated with BCB staining, followed by NMB. In addition, the senior observer exhibited a higher bias in counting compared with their junior counterpart. Repeated counting also correlated with a higher rate of bias. Nevertheless, inter-observer consistency was high (intraclass correlation coefficient >0.95) and inter-/intra-observer variations were non-significant (both *p* > 0.05). Our results supported the use of BCB stain for reticulocyte enumeration and the reliability of manual counting despite the involvement of human factors, which had negligible impacts on the final outcomes.

## 1. Introduction

Reticulocytes are non-nucleated, immature erythrocytes produced from their precursors in the bone marrow during the process of erythropoiesis in which enucleation occurs before being released into the circulation in billions on a daily basis about one day before maturation [1,2,3]. During the mechanosensitive ion channel PIEZO1-mediated maturation process [4], reticulocytes reduce their plasma membrane by 20% and remove their residual internal organelles through lysosomal protein degradation, autophagy, vesiculation [5], and cytoskeleton rearrangement [6]. In the maturation process, reticulocytes also release exosomes that have been found to mediate intercellular communications and may also play a diagnostic role in viral infections (e.g., COVID-19) [7]. Their characteristic network of ribosomal RNA gives them their reticulo-filamentous appearance [5].

The use of an automated hematology analyzer can give an absolute reticulocyte count [8]. Furthermore, the use of a complete blood cell counter (e.g., Sysmex XE-5000) in which the reticulocytes are labeled with polymethine or other fluorescent dyes for automatic quantification based on fluorescent intensity [9,10] can provide important information, such as immature reticulocyte fraction, mean reticulocyte cell volume, and reticulocyte hemoglobin content. Nevertheless, despite its accuracy, the costs of the equipment and procedures remain an obstacle to its popularity in clinical application [11]. Another even more expensive and procedurally complicated approach to reticulocyte counting that shares the same problems with popular clinical applications is flow cytometry. However, it has been reported to offer significant advantages over manual counts [12]. In addition, the presence of nucleated red blood cells (RBCs), Howell–Jolly bodies, sickle cells, or giant platelets may impair the accuracy of flow cytometric reticulocyte enumeration [12]. Therefore, the manual counting of reticulocytes by light microscopy with supravital dyes for RNA, which was first introduced in the 1940s, remains the standard means of reticulocyte enumeration [8].

Despite its routine clinical use, the accuracy of manual counting is affected by the staining technique, stability of the dyes, resolution of the microscope, red blood cell count, evenness of blood cell distribution on blood smears, as well as human factors [10,13]. There are two common supravital dyes for reticulocytes for manual microscopic counting, namely, brilliant cresyl blue (BCB) and new methylene blue (NMB), both of which target the ribonucleic acid residues in reticulocytes and confer them the characteristic blue reticular staining pattern. Liu’s stain (i.e., an improved staining approach based on the Romanosky stain [14]) is also commonly used for fixation because of its methanol or ethanol component [15]. The current study aimed at elucidating the impacts of the selection of staining methods and human factors (i.e., seniority) on the accuracy of manual reticulocyte enumeration.

## 2. Materials and Methods

### 2.1. Study Design and Protocol

A total of 100 consecutive 3 mL venous blood samples acquired through conventional venipuncture were collected in 5 mL glass tubes with ethylenediaminetetraacetic acid (EDTA) (BD Vacutainer^®^ blood collection tubes, Franklin Lakes, NJ, USA) from 100 patients subjected to reticulocyte enumeration at a tertiary referral hospital. Four blood smears were made from each sample, each of which was stained with one of four staining methods (i.e., BCB, NMB, BCB with Liu’s stain, and NMB with Liu’s stain). Two laboratory technologists, namely, one senior (i.e., working experience over 20 years) and one junior (i.e., working experience less than three years), took turns in performing reticulocyte counting on the four slides from each blood sample (Figure 1).

With the results of reticulocyte count from an automated system (Sysmex XE-5000) serving as the standard for comparison, the accuracy of the four staining approaches was compared. In addition, by categorizing the results of Sysmex XE-5000 into three ranges, namely, normal (0.1–2.0%), high (2.1–5.0%) and extremely high (5.1–18.0%), we investigated whether the magnitude of reticulocyte counting could affect the accuracy of any of the staining methods. Furthermore, the impacts of human factors (i.e., seniority, inter- and intra-observer variations) on the accuracy of enumeration were evaluated.

### 2.2. Participants and Blood Sampling

All blood samples were disconnected from the identity of the subjects from which they were collected. Coagulated samples or those with hemolysis were excluded from the present study. The protocol and procedures of the study were reviewed and approved by the institutional review board (IRB) of the Chang Gung Medical Foundation (Approval no. 201600484B0). All participants signed an informed consent form before participating in the present study. 

### 2.3. Automated Reticulocyte Enumeration with Sysmex XE-5000

Reticulocyte count from nucleic acid fluorescence staining with polymethine using the Sysmex XE-5000 system (Kobe, Japan) served as the standard for the current study. The procedures for counting were in accordance with the manufacturer’s instructions. Briefly, the system used 130 μL of whole blood from each of the 100 samples. The blood samples then underwent the counting process after being automatically stained with polymethine.

### 2.4. Procedures of Reticulocyte Count with Manual Method

The reticulocyte count through conventional microscopy was performed according to a standard protocol. To each aliquot of 200 μL whole blood in a polystyrene tube, 200 μL of either 1% BCB or 1% NMB was added. The mixture was homogenized before being placed inside a light-shielded container for 15 min at room temperature. A drop of the mixture (about 25 μL) was then transferred to a 26 mm × 76 mm glass slide for preparing a blood smear. For each blood sample, four blood smears were prepared. While two blood smears were stained with BCB, the other two were stained with NMB. After drying at room temperature, one of the smears stained with BCB and another stained with NMB further underwent Liu’s staining procedure. Therefore, for each blood sample, four blood smears were prepared, namely, BCB, BCB with Liu’s stain, NMB, and NMB with Liu’s stain (Figure 1).

Reticulocyte counting was conducted under a microscope with 1000× immersion objective lenses on a disc square. For each blood smear, 20 microscopic fields were observed. Because each field contained approximately 50 blood cells, a total of over 1000 erythrocytes were examined for each blood smear (Figure 1). The number of reticulocytes was then calculated, with the results being expressed as reticulocyte percentages [10]. Each observer then repeated the procedure for each slide once, with the final reticulocyte count being the average of the results from the two counting procedures. For comparison with the result from Sysmex XE-5000, the average counting from the two observers for each staining technique was used. Typical blood smears stained with the four methods for reticulocyte enumeration are shown in Figure 2.

### 2.5. Statistical Analysis

All statistical analyses were conducted with the statistical software package SPSS (SPSS Inc., Chicago, IL, USA). A box plot was used to compare the pattern of data distribution among the results of reticulocyte enumeration using the four staining techniques, as well as the discrepancies between their mean/median values and the automated standards (i.e., Sysmex XE-5000). For each blood sample, the deviation (i.e., bias) of the results of each manual counting method from the standard value was numerically assessed with the equation (y − x)^2^ (where x: result from automated enumeration; y: average value of manual reticulocyte count from the two observers). The average bias in enumeration between the results from a specific staining technique and those from the automated approach is expressed as ∑(y − x)^2^/n (where n: total number of samples). ANOVA was used to investigate the potential impact of the three categories of machine-acquired data (i.e., normal, high, extremely high) on the accuracy of manual enumeration. Intraclass correlation coefficient (ICC) was adopted for determining the consistency of results between the two observers, with an ICC > 0.75 being defined as a satisfactory consistency. A two-way *t*-test was used to assess the significance of the difference in the result of reticulocyte enumeration between the two observers (i.e., inter-observer variation) as well as the consistency of data reproduction for each observer (i.e., intra-observer variation). A probability value (*p*) less than 0.05 is considered statistically significant.

## 3. Results

### 3.1. Correlation between the Results of Automated Reticulocyte Enumeration and Manual Counting with Four Staining Approaches

The results of the box plot analysis of data distribution for the four manual counting approaches are shown in Figure 3. BCB exhibited a mean/median count closest to the result from automated enumeration with the narrowest inter-quartile ranges among the four methods, followed by NMB, which also demonstrated a mean/median count closer to the result of automated enumeration than that of the other two approaches (i.e., BCB with Liu’s stain and NMB with Liu’s stain). Consistently, examination of the discrepancy in result between a specific staining technique and the automated approach (i.e., mean bias) through mathematical amplification (i.e., squaring of the difference) identified BCB as the method that gave a reticulocyte count closest to the standard value (Table 1). The accuracy was followed by NMB, BCB with Liu’s stain, and NMB with Liu’s stain.

### 3.2. Impact of Value of Reticulocyte Count on Accuracy of Enumeration Using Different Staining Approaches

ANOVA demonstrated that the value of reticulocyte enumeration [i.e., normal (0.1–2.0%), high (2.1–5.0%) and extremely high (5.1–18.0%)] had no significant influence on the accuracy of counting (i.e., difference in result between manual and automated counting) (all *p* > 0.05).

### 3.3. Inter-Observer Variation

Interestingly, the overall bias of the senior technologist appeared to be consistently higher than that of the junior member (Figure 4). Nevertheless, examination of the intraclass correlation coefficient (ICC) showed a high degree of consistency in the result of enumeration between the two observers for each sample regardless of the staining technique (all ICC > 0.95) without significant inter-observer bias (all *p* > 0.05) (Table 1). 

Another intriguing finding was that despite the overall similar pattern of stain-related bias noted during the first and second counting for both observers, the senior technologist tended to produce more bias in counting from blood smears using BCB with Liu’s stain compared to the corresponding result of the junior technologist (Figure 4).

Regarding the impact of repeated counting, there was a small but consistent increase in deviations from the standard values from all four staining techniques for both observers when conducting the second counting procedure compared with the first one (Figure 4).

### 3.4. Intra-Observer Variation

The two-way *t*-test for assessing the consistency of data reproduction for each observer demonstrated non-significant variation between the results from the first and second manual enumeration (both *p* > 0.05) (Table 1), suggesting negligible intra-observer bias.

## 4. Discussion

The enumeration of reticulocyte in peripheral blood is critical for the assessment of functional integrity of the bone marrow. In addition to being one of the keys to the identification of the causes of anemia [16,17], a reticulocyte count is a diagnostic tool for a variety of diseases when combined with other laboratory parameters. For instance, anemia with a low reticulocyte count was found to be a tell-tale sign in the diagnosis of parvovirus B19 infection for recipients of solid organ transplantation [18]. Moreover, an elevated inter-twin difference in hemoglobin concentration together with a large inter-twin reticulocyte count ratio is an important neonatal hematological diagnostic feature of anemia-polycythemia sequence (TAPS) in monochorionic twins [19,20]. In addition to being a diagnostic tool, reticulocyte enumeration serves as a reliable indicator of therapeutic response. While a reduction in reticulocyte counts with an increased hematocrit indicates a favorable treatment response for patients with sickle cell disease [21,22], an increase in both reticulocyte counts and hemoglobin level signifies a clinical improvement in patients diagnosed with parvovirus B19-associated pure red cell aplasia after kidney transplantation [23]. Furthermore, the reticulocyte count can play a therapeutic role as it helps in guiding the decision of blood transfusion for extremely low birth weight infants [24]. Nevertheless, reticulocyte enumeration has its diagnostic limitations. For instance, reticulocyte production index (RPI) may not be an adequate tool for assessing bone marrow erythropoietic capacity in children [25]. 

The normal reticulocyte count in healthy individuals ranges from 20 to 110 × 10^9^/L, depending on the method of counting [26]. The reference ranges for reticulocyte count vary with age. While the reticulocyte count in healthy newborns ranges from 2.5% to as high as 6.5%, it is normally between 0.5% and 2% in children and adults [27]. An abnormal reticulocyte count is a common diagnostic clue to a number of erythropoiesis-related conditions. While an increased reticulocyte count represents ongoing or recent erythrocyte production in response to situations such as hemorrhage or hemolysis, a decreased count reflects a decreased erythrocyte production, including anemia from iron or vitamin deficiency (e.g., folic acid, vitamin B-12) as well as suppressed bone marrow function attributed to radiation therapy, diseases such as aplastic anemia, metabolic storage disorders, leukemias, or sarcoidosis as well as pathogen-related causes such as infection [28]. Reticulocyte enumeration is especially of value in monitoring the regenerative activity of bone marrow following chemotherapy or bone marrow transplantation [8].

Despite the accuracy of the automated reticulocyte enumeration, the cost of equipment and the complex procedure restrict its wide clinical application [11]. In addition, updated evidence has shown that automated reticulocyte enumeration does not render manual counting obsolete. Not only did a recent investigation demonstrate comparable accuracy in reticulocyte enumeration between the automated approach using the Sysmex XT-2000i analyzer and manual counting on freshly prepared (i.e., less than six hours) blood smears [29], but a previous animal study also reported the need for manual verification because of malaria-related pseudoreticulocytosis from the automated approach [30]. Therefore, manual reticulocyte counting using supravital dyes remains the gold standard for reticulocyte enumeration. The current study represented the first investigation into the effect of the choice of staining technique on the accuracy of manual reticulocyte enumeration as well as the potential impact of human factors in the process. Our results not only identified BCB as the staining method that gave an enumeration result closest to that acquired through the automated approach but also supported the accuracy of manual reticulocyte counting without significant adverse influence from human factors.

Although both BCB and NMB are supravital dyes targeting the ribosomal RNA in reticulocytes for microscopic enumeration, their effects on the accuracy of counting remained unclear. In addition, the influence of fixation with Liu’s stain on the results of enumeration was not addressed. The present study not only identified BCB as the most reliable staining approach to reticulocyte enumeration compared with the other three methods but also found that Liu’s stain was associated with individual-dependent bias in reticulocyte enumeration. The adverse impact of Liu’s stain on the counting process, which could probably be explained by individual variations in interpretation of a positive staining, may raise questions regarding the necessity of this additional fixation procedure.

Another important finding of the current study was that the result of reticulocyte enumeration (i.e., normal, high, and extremely high) did not affect the accuracy of manual counting. Therefore, our results ruled out the impact of reticulocytosis (i.e., an increased number of peripheral blood reticulocytes as in anemic patients with the functional integrity of bone marrow) [8] on the correct interpretation of the result of reticulocyte enumeration.

Focusing on the potential influence of human factors on the accuracy of reticulocyte counting, the current study revealed that the seniority (or age) of the observer, the application of a certain staining technique, and a repeated procedure could have an impact on the accuracy of reticulocyte enumeration. Regarding our finding of a negative albeit non-significant influence of the seniority of the observer on the counting results, an age-related reduction in the sharpness of observation in a routine procedure that requires relatively little experiential input for interpretation may be an explanation. In respect of the potential benefit of repeated counting, although the two technologists were aware of the experiment, there was no change in the whole protocol from the staining of a blood smear to microscopic reticulocyte enumeration. The only extra effort that was made to enhance accuracy and consistency was repeating the counting procedure for each blood smear which they performed just once in actual clinical practice. Interestingly, our study showed that such an extra effort did not enhance the accuracy of reticulocyte enumeration. Therefore, our findings suggested that seniority (or age) and fatigue during second counting may adversely affect the result of reticulocyte enumeration. In this way, our study highlighted the existence of human factors in the enumeration process. Nevertheless, the impacts were not statistically significant, and the results of counting showed a high consistency both between the two observers and between the two counting procedures for each observer, supporting the reliability of the manual counting approach.

## 5. Conclusions

The current study showed that BCB staining was associated with the least deviation from the result of automated counting. Moreover, inter-observer consistency was high, and both inter-/intra-observer variations were non-significant. In addition, although seniority and repeated counting may be related to an increased risk of counting bias, they had no significant impact on result consistency.

There were several limitations in the current study. First, because the data for analysis were disconnected from the patients’ demographic and clinical information, whether the demographic and disease characteristics of the patients have an impact on the quality of staining (e.g., intensity) remain unknown. Second, the small number of samples with reticulocytopenia (i.e., a decreased number of reticulocytes) precluded our investigation into its potential effect on the accuracy of reticulocyte enumeration. Third, data from the flow cytometry were not available for comparison.

## Figures and Tables

**Figure 1 diagnostics-12-02154-f001:**
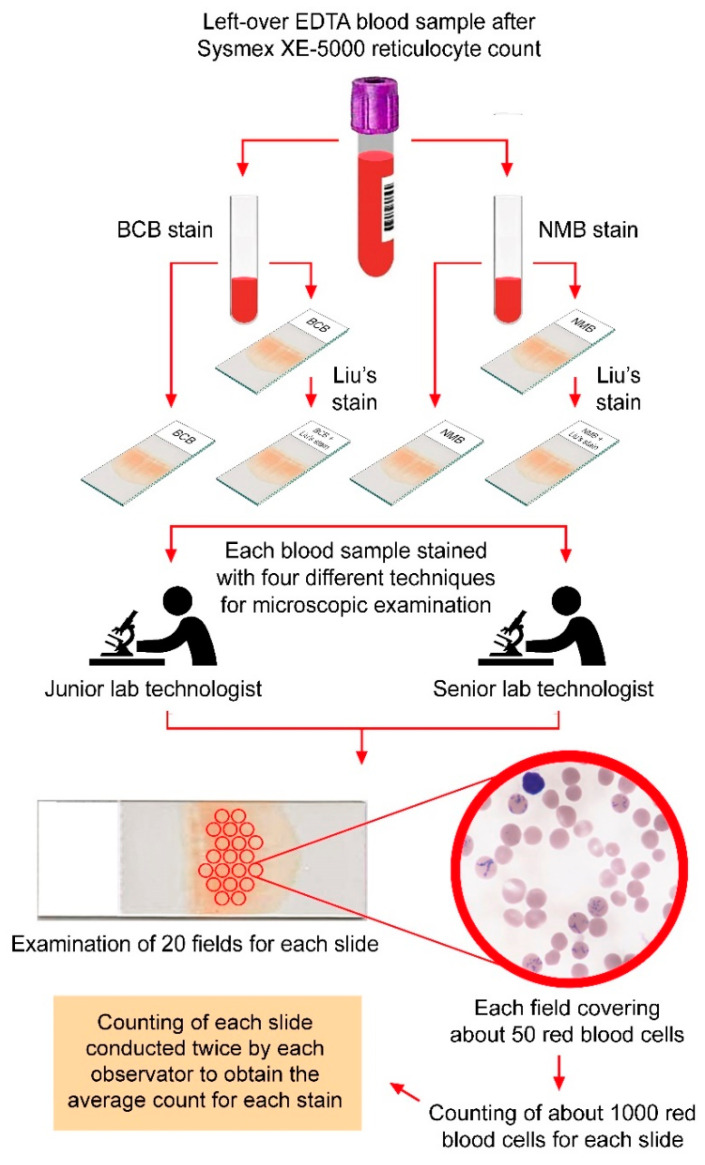
Protocol of microscopic reticulocyte enumeration conducted by two observers using blood smear stained with four different methods. BCB—brilliant cresyl blue; BCB + LIU—BCB with Liu’s stain; EDTA—ethylenediaminetetraacetic acid; NMB—new methylene blue; NMB + LIU—NMB with Liu’s stain.

**Figure 2 diagnostics-12-02154-f002:**
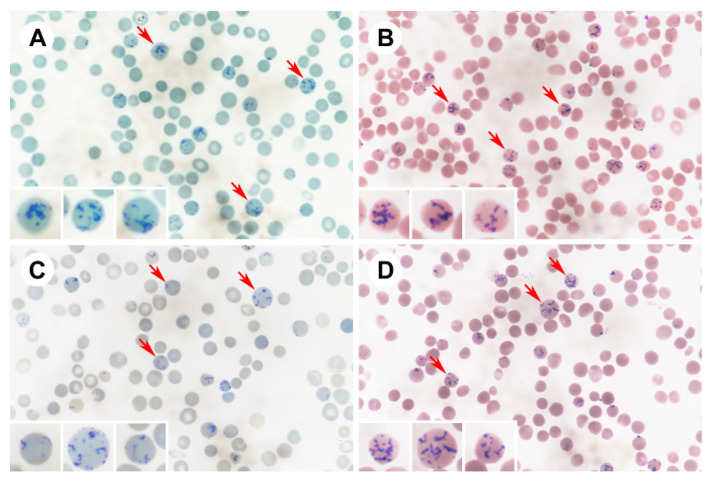
Blood smears processed with (**A**) brilliant cresyl blue (BCB), (**B**) BCB and Liu’s stain, (**C**) new methylene blue (NMB), and (**D**) NMB with Liu’s stain being microscopically examined with 1000× immersion objective lenses. Representative reticulocytes labeled with arrows. Note the characteristic blue reticular staining pattern of ribonucleic acid residues (lower-left corners).

**Figure 3 diagnostics-12-02154-f003:**
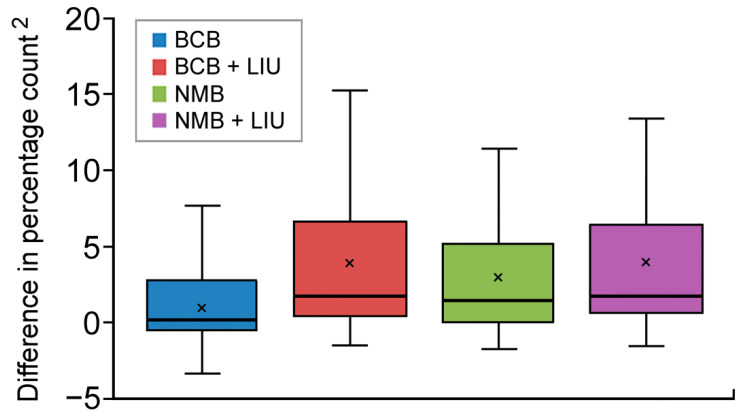
Box plot showing the distribution of differences in reticulocyte counting between the automated analyzer and four different supravital staining approaches, showing the minimum and maximum values as well as the first (lower), second (median), and third (upper) quartiles. Small crosses denoting mean values. BCB—brilliant cresyl blue; NMB—new methylene blue; LIU—Liu’s stain.

**Figure 4 diagnostics-12-02154-f004:**
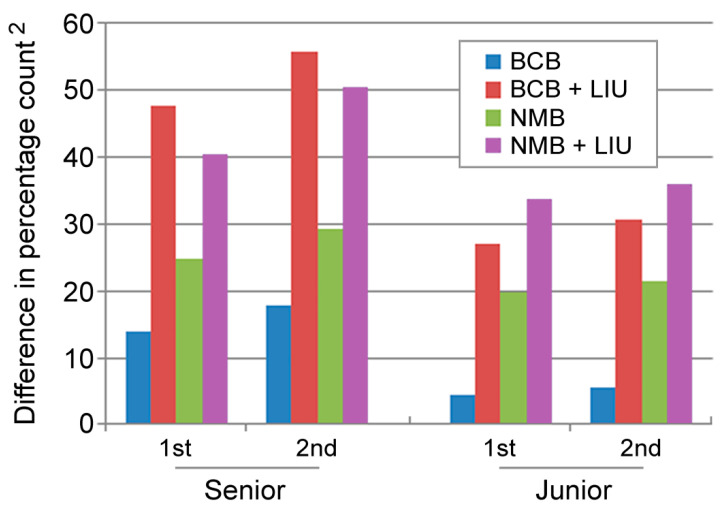
Variations in reticulocyte enumeration bias with staining method, observer, and repetition of counting. BCB—brilliant cresyl blue; NMB—new methylene blue; LIU—Liu’s stain.

**Table 1 diagnostics-12-02154-t001:** Bias in reticulocyte enumeration with four staining methods compared with automated counting and inter-/intra-observer variations.

Staining Method	Mean Bias ^*a*^	Inter-Observer Variation		Intra-Observer Variation (*p*) *^c^*	
		ICC ^*b*^	*p ^c^*	Senior	Junior
BCB	8.77	0.95	0.324	0.725	0.784
BCB + LIU	38.8	0.972	0.395	0.681	0.782
NMB	23.29	0.99	0.687	0.716	0.881
NMB + LIU	39.38	0.988	0.689	0.61	0.807

BCB—brilliant cresyl blue; NMB—new methylene blue; BCB + LIU—BCB with Liu’s stain; NMB + LIU—NMB with Liu’s stain; *^a^* Mean bias assessed with the equation ∑(y − x)^2^/n (where x—value from automated enumeration; y—average value of manual reticulocyte count from the two observers; n—total number of samples); *^b^* ICC—Intraclass correlation coefficient with >0.75 being defined as satisfactory consistency; *^c^* Significance of difference determined with a two-way *t*-test.

## Data Availability

Not applicable.

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
