# Peer review of "Impact of Staining Methods and Human Factors on Accuracy of Manual Reticulocyte Enumeration"

_diagnostics, 2022, doi:10.3390/diagnostics12092154_

Round 1

Reviewer 1 Report

The authors stained hematological slides containing reticulocytes with different stains in order to compare their usefulness and also checked the difference in reliability between machine counting and human counting.

Specific Points of Criticism and Suggestions for Alterations:

(1)  Title:  „Impact“ should be used (instead of „Impacts“).

(2)  Discussion:  It appears that the humans achieved similar results as machine counting. Were the two technicians aware of the experiment, maybe making an extra effort for accuracy and consistency (thus achieving a degree of reliability which cannot be reached in the daily routine)? This issue may be mentioned and/or discussed in the section Discussion.

(3)  Figure 2:  Maybe add to each of the four photos an inset in a corner of each picture showing one or more positive cells at larger magnification.

(4)  Line 195:  How to explain the higher bias of the senior technician?

Reviewer 2 Report

This is a very interesting paper reporting very important clinical findings. I have no major concerns. Authors have a knowledges all limitations and decently approached the issue. 
